# Analysis and Review of the Research and Advocacy for Behavioral Change Related to the Denormalization of Gender Violence in Spanish Universities

**DOI:** 10.3390/bs15040500

**Published:** 2025-04-09

**Authors:** Marta Soler-Gallart, Mar Joanpere, Lidia Bordanoba-Gallego

**Affiliations:** 1Department of Sociology, University of Barcelona, 08034 Barcelona, Spain; lidiabordanoba@ub.edu; 2Department of Business Management, Universitat Rovira i Virgili, 43204 Reus, Spain; mar.joanpere@urv.cat

**Keywords:** gender violence, Spanish university, supportive network, safety, history

## Abstract

Research on gender violence in Spanish universities began in 2003, 8 years after the first official denunciation. Conducting the first statewide survey on this issue was a significant step. This study provided essential data on gender violence in these institutions and reviewed effective prevention and response strategies from other countries. Further qualitative studies emerged that analyzed the behaviors of faculty, students, staff, decision-makers, and the media, which either perpetuated or prevented gender violence, along with the psychological and health impacts on victims and their supporters. After more than 20 years, a comprehensive literature review is needed to systematize these findings. To address this gap, a literature review was conducted to examine the behavioral changes within the university community and other relevant social actors regarding gender violence in universities. The results indicate that, although there is still some resistance, significant behavioral shifts have occurred, fostering a supportive network among faculty, researchers, staff, and students, which has contributed to an increased sense of safety.

## 1. Introduction

Gender-Based Violence (GBV) represents one of the most persistent systemic issues in our society, directly affecting the lives of nearly 736 million women—almost one in three women—worldwide ([31]). This issue is also recognized within Sustainable Development Goal 5: Achieve gender equality and empower all women and girls, specifically in target 5.2, which states the following: “Eliminate all forms of violence against all women and girls in the public and private spheres, including trafficking and sexual and other types of exploitation”.

Gender-based violence is a problem that affects multiple sectors globally ([39]), underscoring the need to study it in depth across various contexts. Its presence also extends to the university environment worldwide ([30]), where it causes physical and mental health problems for victims ([17]; [37]).

Research on gender-based violence in Spanish universities began in the early 2000s, with significant preparatory work starting in 2003. It was officially funded in 2005, marking the beginning of the first official project on the subject. This was conducted between 2005 and 2008 through a project titled “Gender-Based Violence in Spanish Universities” ([33]). This I+D+I (Research, Development, and Innovation) project, led by Rosa Valls, was coordinated by the research group Community of Research on Excellence for All (CREA) and funded by the Spanish Women’s Institute. Until then, there were no specific studies addressing this phenomenon in Spanish universities, making this initiative the first national survey dedicated to the topic ([35]). The project is part of a research line on gender-based violence previously undertaken by researchers from the SAPPHO Women’s Group of CREA. Prior to these studies, research had already been conducted on the presence of gender-based violence in universities in countries such as the United States, Canada, and the United Kingdom ([35]). These precedents highlight the importance of studying this phenomenon in the Spanish context, demonstrating that, as a global issue, its analysis in Spanish universities is equally essential for understanding and addressing its specific impact.

One of the most significant findings from the initial research was that 65% of respondents reported having experienced or been aware of situations of gender-based violence within the university context ([33]). This finding, along with others from this study, helped to shed light on the issue of gender-based violence through concrete data ([6]).

Support networks are critical in creating violence-free spaces, as fostering an environment conducive to reporting encourages victims to feel supported when filing complaints ([8]; [10]). These networks should not only focus on providing support but also on developing mechanisms to assist victims ([26]), as they contribute positively to mental health by mitigating the effects of potential traumatic stress that victims may experience ([27]). Post-traumatic stress generates a continuous state of hyperarousal and difficulties in regulating emotions in reaction to environmental cues ([21]).

Additionally, gender-based violence can produce severe mental health consequences due to the traumatic nature of such experiences, both for those directly affected and for those who provide support ([38]; [2]). For victims, witnessing the individuals supporting them face violence creates a dual process of revictimization, exacerbating anxiety and distress ([2]). The attacks suffered by those who support victims are referred to as Isolating Gender-Based Violence ([18]).

It is also important to highlight that in socialization processes, unequal values are transmitted, which over the course of our lives shape behaviors that perpetuate gender inequalities ([35]). This influence can contribute to the perpetuation of gender-based violence ([19]), as it is closely linked to an attraction toward violent behaviors, identified as one of the main sources of gender-based violence ([25]). This underscores the need to implement preventive interventions focusing on socialization processes ([24]).

The primary purpose of this research is to analyze the relationship between structural changes in Spanish universities and behavioral modifications among the individuals within these institutions in relation to gender-based violence. The significance of this study lies in the review presented, which not only organizes the changes that have occurred over more than 20 years but also analyzes behavioral changes within the university context, as well as the psychological and health impacts on victims and their support networks.

The research question is as follows:

How have the demands of social actors influenced the formulation and implementation of legislative and structural changes on gender-based violence in universities, and what impact have these changes had on attitudes and behaviors?

Thus, the main objective of this study is to analyze transformations in the attitudes and behaviors of the university community, as well as those of other social actors, regarding gender-based violence in the university context.

The key conclusions of the research emphasize that social demands have driven structural and legislative changes that have improved safety and contributed to the denormalization of gender-based violence within universities.

## 2. Materials and Methods

The review conducted in this research was designed based on PRISMA standards ([29]) to ensure the transparency of this study. The process of searching for and selecting articles is detailed in the flow diagram (Figure 1).

The search strategy was carried out using the databases Web of Science (WoS) and Scopus, chosen because they include journals indexed in the most recognized impact factors, such as the SSCI from WoS and the SJR from Scopus. Keywords were searched in the title and keywords sections of the articles in both databases. The keywords used were the following: “Gender violence” and “Spanish universities”; “Feudalism” and “Spanish universities”; “Gender-based violence” and “Spanish university”; “Isolating Gender Violence” and “universities”; “Gender violence” and “female university students”; “Gender violence” and “Isolating Gender Violence”; “Sexual harassment” and “solidarity networks”.

The search identified a total of 104 articles, 50 in WoS and 54 in Scopus.

The inclusion and exclusion criteria for the articles were as follows:-The articles address gender-based violence in Spanish universities.-The articles analyze social or structural changes in Spanish universities related to gender-based violence.

The analyzed articles were published in peer-reviewed journals across various fields, including health and psychology. Additionally, all the included articles were written in English.

Similarly, to conduct a more comprehensive study on Behavioral Change in the Spanish university community, various sources have been consulted, including the original laws and the CREA chronology ([9]). This is because the most significant advances have been driven by this research group, as evidenced in the results section.

## 3. Results

### 3.1. Breaking the Silence and Attacks

The first attempt to address gender-based violence in Spanish universities took place in 1995 under the leadership of Professor Ramon Flecha from the University of Barcelona, who was then the director of the Community of Research on Excellence for All (CREA) ([23]; [28]). In this initial effort, Ramon Flecha approached the university’s governing council to emphasize the need to address gender-based violence in universities. Flecha argued that his strategy was inspired by excellence practices implemented by leading universities globally and proposed the establishment of a commission to oversee activities related to this issue ([23]).

The governing council never provided an official response to this request. Furthermore, in 1999, faculty members joined the fight against gender-based violence in universities with the formalization of SAPPHO, the women’s group within CREA ([23]). This demonstrated the persistent silencing of the issue by institutional authorities. The creation of this group was the result of a long-standing effort by various members of the CREA research team, who had been investigating and publishing on gender issues.

Subsequently, in 2002, the SAPPHO women’s group from CREA submitted a research project on gender-based violence in universities but failed to secure funding ([23]). It is worth noting that between 1983 and 2005, none of the 6,955 I+D+I projects funded by the Women’s Institute or the Ministry of Education and Culture addressed gender-based violence in universities ([23]).

However, in 2003, research on gender-based violence in Spanish universities began. Simultaneously, in 2003 in Catalonia, the Unitary Platform Against Gender-Based Violence was established under the slogan “Trenquem el silenci” (“Break the silence”). This represented a change as, for the first time, members of the university community addressed the problem courageously, despite the risks involved. As previously evidenced, both the institution and its members had maintained a long silence in the face of gender-based violence. This platform was created in collaboration with the SAPPHO women’s group, composed of individuals from the university community. As a university-based group, they decided to break the silence around gender-based violence in universities. Thus, in 2003, the first research on gender-based violence in Spanish universities began. That same year, CREA proposed the formation of equality commissions and decided to publicly address the issue of gender-based violence, both within universities and within CREA itself ([28]).

In 2004, the research on gender-based violence in Spanish universities was submitted for an I+D+I project call. That same year, Spain passed the first Gender Violence Organic Law in Europe, Organic Law 1/2004. Following the approval of this law, the conference “Against Harassment, Zero Tolerance” was held at the Science Park, organized by the Platform Against Gender-Based Violence. During this event, Esther Oliver presented a book on gender-based violence. Discussions during the conference highlighted a critical flaw in the new law. Political leaders and CREA researchers debated the prior work developed within the research group to prevent the socialization of gender-based violence. These discussions created opportunities to address the issue in the context of Spanish universities ([23]).

Between 2005 and 2008, a breakthrough occurred, as the Women’s Institute funded the I+D+I project on gender-based violence in Spanish universities under the title ‘Gender-Based Violence in Spanish Universities’, directed by Rosa Valls and coordinated by the CREA research group ([33]). This project was the first to investigate gender-based violence in Spanish universities, making sexual harassment in universities visible and recognizing this problem within the academic sphere ([23]).

The research revealed a lack of recognition of gender-based violence incidents and formal complaints within universities. This was partly due to the normalization and tolerance of violence, as well as the feudal structure of universities, which suppressed reporting behaviors in cases of gender-based violence ([34]; [6]). The feudal university system refers to a rigid and hierarchical structure ([1]), based on relations of abuse of power and bullying dynamics ([12]).

However, gender-based violence was not only linked to institutional factors but also to coercive discourse that associated attraction with violent men, thereby diminishing the appeal of men with egalitarian attitudes. This dynamic perpetuated gender-based violence and had significant effects on victims’ health ([25]).

In 2007, this project had a political impact that changed Spanish universities. Members of the Spanish parliament belonging to diverse ideological orientations had dialogues with the members of CREA about their research and how to implement the findings politically. The first impact was that the Parliament approved legislation making it mandatory for universities to establish Equality Units and implement protocols to prevent and address sexual harassment ([3]). The second impact was the change in the feudal process to select and promote university professors to a meritocratic one ([4]).

This provided victims of harassment with a safer space to report their situations ([6]). This commitment culminated in the approval of Organic Law 4/2007 on 12 April 2007, which brought about significant improvements in addressing gender-based violence in Spanish universities ([23]; [6]).

Following this organic law, universities began to develop specific protocols against gender-based violence. Since 2007, they have been required to establish Equality Units and implement preventive measures. For the first time, mechanisms were put in place to allow affected individuals to report their situations, effectively breaking the “law of silence” that had previously prevailed, even among academic staff who were aware of such cases ([23]). Furthermore, the governmental agency disseminated the study’s results, generating significant media and societal impact and fostering a consensus on the need to address gender-based violence in Spanish universities ([23]).

However, within the university environment, led by the CRUE (Conference of Rectors of Spanish Universities), there was not only a refusal to acknowledge the existence of this problem but also an active effort to silence it, arguing that exposing such issues would harm the university’s reputation. Some individuals who broke the silence were even threatened with expulsion in an attempt to dissuade others from speaking out ([6]).

This response stemmed from the feudal system of faculty selection ([15]). In Spanish universities, the feudal system was characterized by a hierarchical structure in which senior professors, such as full professors, wield significant power over crucial decisions, including faculty hiring. This system encouraged dynamics of submission and perpetuated impunity regarding sexual harassment and gender-based violence within universities. Attempts to report or recognize such issues often resulted in retaliation against those who “broke the silence”. These responses not only sought to punish those who dared to speak out but also served as a warning to others, reinforcing a culture of impunity that allowed gender-based violence to persist within universities. The feudal system effectively guaranteed the impunity of sexual harassment, mediocrity, and hypocrisy ([15]; [6]).

In response to this situation, six months later, on 5 October 2007, accreditation for access to university teaching positions was introduced through Royal Decree 1312/2007 ([4]; [6]; [15]). As outlined in this decree, which was based on the reforms introduced by Organic Law 4/2007 of 12 April, one of the key aspects was the implementation of a national accreditation system. This system required candidates for university teaching positions to be nationally accredited beforehand. The research team behind the I+D+I project that inspired these two reforms ([34]) argued that the reforms were interdependent and that one could not be implemented without the other ([6]).

With the enactment of this reform, universities transitioned from a feudal system characterized by an abusive and hierarchical structure to a meritocratic system based on national accreditation. These laws drove democratic reforms in universities, promoting merit-based selection processes, combating sexual harassment within universities, and fostering gender equality as a critical factor for improving academic productivity ([6]).

After more than a decade of fighting against gender-based violence in universities, two opposing attitudes are evident. On the one hand, those who decided to make visible a hitherto hidden problem, assuming the risks that this entailed. On the other hand, the conduct of the aggressors and their accomplices, who tried to prevent any denunciation of gender violence in the university environment, which had been silenced and normalized through coercion and threats. This opposition of attitudes not only influenced the progress of the fight against gender-based violence in universities but also had a significant impact on the health of those who faced the problem, both on the victims themselves and on those who supported them ([2]).

### 3.2. Empowerment of Victims

Following the reforms implemented through the project on gender-based violence in universities, and after its completion in 2008, on 25 November 2009, Professor Lidia Puigvert from the University of Barcelona, a member of CREA, sent an official letter to the Dean of the Faculty of Economics and Business. In the letter, Puigvert highlighted that the structure of the master’s and doctoral programs promoted practices that facilitated sexual harassment. Furthermore, the letter detailed the actions of a professor who harassed several female students annually ([15]). That same year, Sarah Rankin, Director of the Office of Sexual Assault Prevention at Harvard, made the following statement about CREA: “Future generations of students and faculty will undoubtedly have a very different experience because of your work” ([15]).

On 26 September 2011, Ramon Flecha reported to the University of Barcelona that her student was being harassed by a professor. The student did not dare to denounce and asked Flecha to do it. The complaint was submitted both to the Equality Commission of the Faculty of Economics and Business at the University of Barcelona and to the Office for Sexual Assault Prevention and Response at Harvard, as the professor’s email indicated he had ties to Harvard. On 5 October 2011, the Equality Commission at the University of Barcelona decided to dismiss the case. On 19 October 2011, the UB received the letter from Harvard and decided to open the case on 25 October 2011. This demonstrates how external pressures, particularly from Harvard, influenced the behavior of the University of Barcelona, leading it to reconsider its position and reopen a case that was initially closed without being investigated. On 13 November 2011, members of CREA collected testimonies from 13 victims. On 5 October 2012, the university referred the case to the Public Prosecutor’s Office, as it could not take internal measures. The only member of Igualdad who supported the victim and drafted the first protocol against sexual harassment for that commission was excluded from it on 4 November 2011 ([15]). Almost a year later, on 5 October 2012, the University of Barcelona sent the case to the Prosecutor’s Office.

In 2013, the Solidarity Network for Victims of Gender-Based Violence in Universities was officially recognized, coinciding with the creation of EROC (End Rape On Campus) in the United States in the same year. This network originated from previous work by the SAPHOO women’s group, as well as efforts and achievements from prior years. That same year, dialogue began with the United States, and the Equality Observatory in Spain recognized the network as a good practice.

The Network broke the silence on harassment in universities, achieving victories that transformed these spaces and behaviors within them. It enabled many more victims to become survivors, transforming universities into violence-free environments ([15]). These violence-free spaces improved the academic environment and helped break the silence ([6]). In 2014, the Solidarity Network for Victims of Gender-Based Violence in Universities, known now as the MeToo University movement, won the first Spanish university sexual harassment case, considered a milestone by MeToo University for reducing impunity. A student was accused of harassment by the victim, who achieved the first successful GBV case in a Spanish university, a master’s student in Sociology at the University of Barcelona in 2014. The victims sought and received support from Professor Ramón Flecha and Professor Marta Soler. Ultimately, the complaint process was successfully concluded that same year, with the student being permitted to continue his studies but prohibited from accessing the facilities and buildings of the University of Barcelona ([15]).

On 5 February 2014, El País, one of the most widely read newspapers in Spain, published statements from the Secretary of Universities of the Generalitat of Catalonia regarding a case of compulsive sexual harassment at the university by the professor previously reported by Flecha about the case of the victim. In his remarks, he emphasized that this unfortunate event would set a precedent. These statements came after the Dean admitted to the Mossos d’Esquadra (Catalan police) that as early as 1987, when she was a student, there were rumors in the faculty about instances of harassment. She explicitly stated, “And it is also regrettable that there are people now saying they knew this had been going on for years and did not report it. And I am not referring to the victims but to those responsible at the university. If they already knew, why didn’t they act?” ([15]). This event marks a shift in behavior, as it illustrates how institutional inaction regarding sexual harassment began to be publicly challenged.

Twenty days after the Secretary of Universities’ statement, on 20 February 2014, students from the Faculty of Economics and Business, along with students and faculty from other departments and victims, held a protest at the rectorate’s office. This change reflects how perceptions of institutional support can spur collective action. Over a thousand individuals signed a petition demanding that the professor, previously reported in 2011, no longer be allowed to teach or conduct research involving students. Ten days later, on 1 March 2014, the Student Assembly of the Faculty of Economics held its first meeting to address the issue of sexual harassment in the faculty. The Solidarity Network for Victims of Gender-Based Violence at Universities was then acknowledged as a Good Practice on 3 March 2014 by the Ministry of Equality-funded Fundación Mujer’s Gender Violence Observatory. Its inclusion in the Database of Good Practices for the Prevention of Gender-Based Violence, which is accessible on the observatory’s website, was one aspect of this recognition ([15]).

The campaign was started on 9 May 2014 by the Solidarity Network and the Unified Platform Against Gender Violence. “Our Daughters Have Rights” ([15]). The following year’s studies were published analyzing experiences of gender-based violence in universities from a behavioral perspective. These works highlighted how the Solidarity Network enabled victims to become survivors, fostering empowerment and resilience. It also emphasized the importance of mutual support and continuous collaboration with organizations dedicated to combating gender-based violence across different sectors of society ([15]). On 3 July 2015, CREA filed a complaint in court to respond to all the anonymous defamations that had been circulated.

In April 2016, discussions began about the reinstatement of the professor who had been expelled two years earlier, with a mandate that he return to teaching. Two months later, on 8 June 2016, a letter from the student movement opposing his reinstatement emerged, successfully preventing his return.

On 13 June 2016, CREA developed the concept of Isolating Gender Violence (IGV). This term refers to attacks and retaliation aimed at those who support victims of gender-based violence, with the goal of isolating them and making it difficult for victims to access the necessary support ([36]; [13]). These health consequences affect both supporters and victims, as witnessing the attacks on the people who have helped them overcome the situation can be even more painful than experiencing the sexual abuse or harassment itself ([2]).

Efforts to combat sexual harassment in universities faced a significant backlash in June 2016 from a group of harassers who launched a massive campaign against CREA, primarily targeting Ramón Flecha, the leader of this transformation ([16]).

The definition of IGV coincided with the day when anonymous accusations were disseminated in the media. Some outlets covered both the statements from the victims’ network and CREA’s position. However, only a small number of media outlets clearly sided with the victims rather than supporting the harassers.

The following day, 14 June 2016, complaints against CREA surfaced, made public through leaked information. As these were not formal accusations and the facts were unproven, CREA never had access to the content of these complaints or knowledge of their origin. It is likely that some of the complainants had never been affiliated with CREA. Five days later, on 19 June 2016, the provincial prosecutor dismissed the complaints under Article 773.2, evidencing the absence of any criminal offense. This demonstrates that, although harassing behavior is still present, its impunity has decreased due to legal support and the gradual elimination of institutional structures that protected the perpetrators, so their efforts no longer prevent complaints.

In 2017, the survivor defended her doctoral dissertation on gender-based violence in universities, which was unanimously approved, after many institutional obstacles. For instance, the Dean and the Equality Commission pressured the Faculty of Economics and Business to reject it. This generated controversy for violating academic standards against a student who had received the Award for Best Undergraduate Student in Sociology and the Award for Best Master’s Student in Sociology ([15]).

In May 2017, Spanish national television (TVE) aired the documentary “Voces contra el silencio” (Voices Against Silence), which addressed the victim’s case. Five months later, in October 2017, the global MeToo movement was founded. Movements such as MeToo University opted for this name as the new designation for the University Gender-Based Violence Victim Support Network, which was initially established in 2013 ([15]). On 16 May 2018, the documentary “Voices Against Silence” Was distinguished with the Global Media Award at the World Media Festival in Hamburg, which represents a relevant achievement for high-quality journalism and the MeToo University movement ([15]). As a documentary aired on Spain’s public television, it broke the silence surrounding various forms of gender-based violence, including violence in universities.

These changes in a decade reflect a transformation in behavior from passivity or complicity to a proactive attitude of making the problem visible. As a result, the balance between the two behaviors mentioned in the previous section, that of aggressors and accomplices, on the one hand, and that of those who denounce violence, on the other, is now tipping towards denunciation and action. This new attitude has led to the creation of the first support networks, increasing the feeling of security in the institutions and encouraging more people to adopt a behavior of active rejection of gender-based violence in universities, so that victims are empowered to take action.

### 3.3. End of Fear and Increasing Willingness to Report

In December 2020, the world’s first legislation recognizing the term Isolated Gender Violence (IGV) was unanimously approved by the Catalan Parliament, four years after the concept was created ([36]). Law 17/2020, issued on 22 December 2008, amended Law 5/2008 on the Right of Women to Avoid Gender Violence. The new legislation became effective on 13 January 2021 ([5]). Following this, other legislatures began drafting their laws addressing IGV ([36]). Additionally, in 2021, the University of Girona introduced the first equality plan by a Spanish university to include Isolating Gender Violence ([32]). In addition to parliaments and universities, numerous entities and even corporations are starting similar proceedings in response to the MeToo University proposal. Organizations like the European Sociological Association (ESA) have also approved their proposal ([15]) in their ethical code of conduct ([11]). It is evident that the laws provide greater support to victims, which translates into greater empowerment and a change in their behavior. By feeling more protected, they have a supportive environment that reinforces their safety and gives them the confidence to report ([7]). Thus, the legal framework provides greater protection for victims, strengthening the victims, isolating the aggressors, and reducing their impunity. In addition, the implementation of these laws helps people who see situations of gender violence to report it, promoting a change of behavior in the defense of victims, and encouraging upstander intervention. The term upstander is used here instead of bystander to highlight the active role of those who choose to intervene rather than remain passive observers ([22]). Moreover, this support contributes to an improvement in their health, thanks to the support received ([2]).

In January 2022, the cover of “El Periódico” featured a single “U”, inside which were the names of 25 survivors of sexual harassment in universities. Journalists reported difficulty finding 25 women willing to appear on the cover, highlighting the pervasive fear within an institution where harassers and IGV remain potent ([15]). Two months later, in March 2022, the Basque Parliament incorporated Isolating Gender Violence into its legislation ([20]). The evidence suggests that legislative support and solidarity networks have played a crucial role in these social changes. The media coverage, such as the front page of *El Periódico*, and the courage of the survivors would not have been possible without this context, which is evidence of a decrease in the impunity of the aggressors.

In September 2022, coinciding with the start of the academic year, during the week of 26–30 September, the MeToo route traveled through 13 Spanish universities in two routes, the northern and the southern, hosting 20 events to support survivors and strengthen solidarity. The initiative fostered awareness and a support network through conferences, discussions, and informal gatherings. Survivors who attended reported improved well-being after connecting with the MeToo community, highlighting the impact of collective support in academia ([7]). This demonstrated the importance of solidarity networks in mitigating the health impacts of gender-based violence and promoting recovery ([7]). Furthermore, the initiative encouraged bystanders to become upstanders in confronting gender violence ([22]).

Throughout 2023, MeToo members organized over 100 public events at various universities across Spain. In 2024, the first conference on objectivity, arbitrariness, and sexual harassment in university faculty evaluations was held at the Institut d’Estudis Catalans. The conference addressed the persistence of sexual harassment in academia and its societal impact. It also highlighted how individuals who support victims help create safer academic environments. Additionally, it underscored how solidarity networks and merit-based, objective evaluations can enhance the academic trajectories of victims. At the end of 2024, a scientific article was published demonstrating that the health of survivors who attended MeToo Route events improved due to the support they received after discovering the MeToo network, as previously mentioned ([7]).

## 4. Discussion

In the results section, the evolution of the response to the issue of gender-based violence in universities over more than twenty years has been presented. It has been shown how, during this period, both individuals within the institution and the institutions themselves have undergone a shift in behavior regarding this issue. This change has been made possible through consistent actions addressing the problem over time.

This change has manifested itself in the two opposing behaviors discussed above. On the one hand, that of those who have decided to speak openly about gender-based violence, breaking the silence of gender-based violence in universities, and on the other hand, that of the aggressors, who have tried to silence these denunciations in order to maintain their impunity and power. However, the analysis shows that the prevailing behavior has been that of the courageous people who have pushed for change. Thanks to their persistence and the implementation of consistent actions aimed at tackling the problem, social and institutional transformation has taken place. This has allowed more and more people to take an active stance against gender-based violence, which in turn has led to legal and structural changes that criminalize the behavior of perpetrators.

One of the most important changes has been the impact on the health of both the victims of GBV and the people who support them. While GBV has very serious consequences, attacks on those who offer support can be even more damaging. Despite this, the support received has been instrumental in reducing harm and improving the well-being of victims.

Focusing on the specific changes that have enabled both behavioral shifts and improvements in health, the following key developments are highlighted. One of the most notable changes since the fight against gender-based violence in universities began is the approach to addressing this issue. What was once avoided and marked by a “code of silence” ([23]) has evolved into a topic openly discussed through public events aimed at fostering reflection on the problem ([7]). These changes occurred without fear of repercussions for victims who report incidents or for those who support them, marking a significant cultural shift. This behavioral change has been driven by legislative and structural changes implemented in recent years, particularly the dismantling of the abusive and hierarchical structure of the feudal system ([34]; [6]).

These legislative and structural changes were prompted by demands for reform. For structural change to occur in addressing gender-based violence in universities, it was essential for individuals within these institutions to change their behavior in response to such situations. Since 1995, this behavioral shift has been crucial, particularly among individuals who challenged normalized behavior in cases of violence. Visibility of the issue also played a key role, starting with the first Spanish project on gender-based violence in universities ([33]).

Additionally, the formation of support networks and safe spaces within universities has been vital. These networks were created through the social support offered to victims by some individuals, despite the reprisals they faced. For example, the support provided by figures such as Ramón Flecha and Marta Soler helped early victims win their cases ([15]). Clear stances against sexual harassment and the significant visibility given to reports and legal proceedings also contributed to the development of the concept of Isolating Gender Violence (IGV). The formal legal recognition of this concept in 2016 allowed for a broader and deeper understanding of gender-based violence and its impacts, particularly on the mental, emotional, and social health of individuals supporting victims ([36]; [13]). IGV has been shown to cause severe health issues ([14]).

The positive impact of victims reporting incidents publicly and the legal backing for those who supported them led to increased safety within universities. Gender-based violence no longer went unpunished, encouraging more victims and witnesses to come forward, knowing they had legal support. Previously, fear of reprisals deterred such actions ([15]; [6]). This development fosters upstander intervention ([22]) and the creation of support networks ([8]; [26]).

This transformation from fear of reporting to proactive action against gender-based violence has improved the academic and scientific environment. By eliminating the hostile climate and breaking the silence, the quality of academic and scientific productivity has also improved ([6]). Thus, the fight against gender-based violence not only enhances individuals’ lives but also boosts scientific productivity.

However, despite significant progress in addressing gender-based violence in universities, resistance and fear of speaking out still persist, as highlighted by the difficulties journalists faced in finding 25 women willing to appear on the cover of “El Periódico” ([15]).

To fully eradicate gender-based violence in Spanish universities, a change in socialization is also necessary. This change should focus on dismantling coercive discourse, which individuals learn through socialization. The attraction to violent behaviors fosters environments where some people adopt behaviors associated with violence to gain acceptance or attraction, sidelining those who promote egalitarian attitudes ([25]). This socialization underscores the urgency of preventive interventions focused on socialization processes.

## 5. Conclusions

The results of this review, responding to the research question, confirm that as changes have occurred in the behavior of faculty, researchers, administrative staff, and students, structural and legislative modifications in the field of gender-based violence have been successfully promoted, thereby improving safety in this environment. Moreover, these transformations have contributed to changing attitudes and behaviors, effectively denormalizing gender-based violence in the university setting. However, future research should further analyze the consequences of each of these changes in greater depth.

It is also important to emphasize that, despite the significant progress made over the past twenty years, continued efforts are necessary to fully establish the university environment as a safe and gender-violence-free space.

## Figures and Tables

**Figure 1 behavsci-15-00500-f001:**
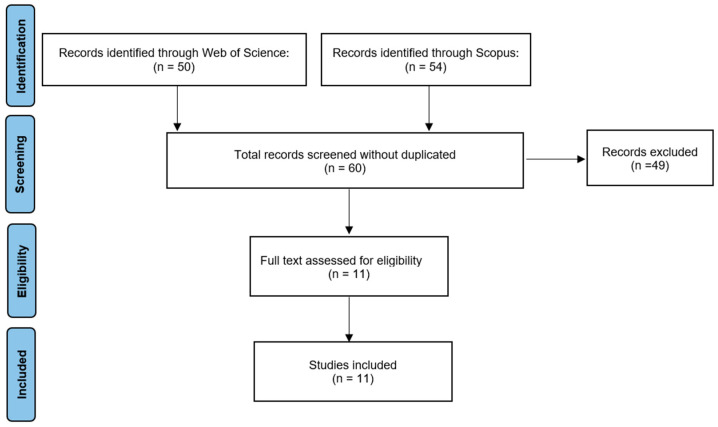
PRISMA flow diagram for the search and screening process.

## Data Availability

Data from this study are available upon request.

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
