# Peer review of "Analysis and Review of the Research and Advocacy for Behavioral Change Related to the Denormalization of Gender Violence in Spanish Universities"

_behavsci, 2025, doi:10.3390/bs15040500_

Round 1
Reviewer 1 Report
Comments and Suggestions for Authors
Thank you for the opportunity to review a very interesting piece of work. It is useful to look at the evolution of university responses to GBV over time. However, I think the title is misleading. It is not a systematic literature review because it does not appraise the research papers. I think it would be better to describe it as a historical overview (or analysis if you include more theory) based on a systematic search of the literature. And I think it would be useful to highlight that it looks at the interaction between structural and behavioural changes rather than how the title currently highlights behavioural change. This change in emphasis is crucial, not just to more accurately describe what the paper is exploring, but also because this relationship is too often overlooked and is a strength of this work. I think this paper could also be strengthened by overtly introducing some theoretical aspects. I am not sure what the most appropriate one would be. Maybe something like social structural theories or ecological systems. I really don't know and don't have time to look anything up at the moment. A large part of this paper is descriptive. You have taken it beyond description in parts, but a clearer theoretical link would strengthen it significantly.
A few more details on the methodology would also be useful to improve trustworthiness.
There were also quite a few small errors, which decreases the reader's trust. For example, lines 25-26, one in three people should be one in three women, although even then the numbers given didn't quite add up. Lines 62-63, I don't think the word hyperactivity on its own is accurate. I think there needs to be another word there according to the reference. A detailed proof read would help.
I+D+I write in full the first time.
Lines 79-80 The aim of the research seems to be more about the relationship between structural and behavioural changes.
Feudalism is not a common term world-wide. It would be good to define its meaning in this context.
State why you have used the term upstander rather than bystander. (I think it is a good choice, but since bystander is more common in some places, an explanation would help.)
Those are my main recommendations. I think it has potential to develop into a stronger paper and is worth the effort.
This work is so important, and I commend you on it. I left a university previously because I reported a male student who intimidated a female colleague and we received virtually no support or protection. The female colleague also left. This was a university with a well known bystander program. The last I heard of the student they were on a court sentencing list. Who knows what they had done and who they had hurt, which may have been prevented if universities were not so scared of litigation. This work is important. That is why I would like to see it strengthened prior to being accepted.
Author Response
MARKS IN GREEN: Revised according to the highlighted similarity parts from the major revision.
- 299-302: We have decided to keep this part because it brings clarity to what is being explained. (Mark in yellow)
MARKS IN BLUE: Reviewers
Title
Comment: However, I think the title is misleading. It is not a systematic literature review because it does not appraise the research papers. I think it would be better to describe it as a historical overview (or analysis if you include more theory) based on a systematic search of the literature. And I think it would be useful to highlight that it looks at the interaction between structural and behavioural changes rather than how the title currently highlights behavioural change. This change in emphasis is crucial, not just to more accurately describe what the paper is exploring, but also because this relationship is too often overlooked and is a strength of this work
Revision: Analysis and Review of the Research and Advocacy for Behavioral Change Related to the Denormalization of Gender Violence in Spanish Universities.
Lines 25-26
Comment: lines 25-26, one in three people should be one in three women, although even then the numbers given didn't quite add up
Revision: directly affecting the lives of nearly 736 million women—almost one in three women—worldwide.
Lines 62-63:
Comment: Lines 62-63, I don't think the word hyperactivity on its own is accurate. I think there needs to be another word there according to the reference. A detailed proof read would help.
Revision: Post-traumatic stress generates continuous state of hyperarousal and difficulties in regulating emotions in reaction.
Line 40:
Comment: I+D+I write in full the first time.
Revision: This I+D+I (Research, Development, and Innovation) project
Line: 79-80:
Comment: The aim of the research seems to be more about the relationship between structural and behavioural changes.
Revision: The primary purpose of this research is to analyze the relationship between structural changes in Spanish universities and behavioural modifications among the individuals within these institutions in relation to gender-based violence.
Line 177-178:
Comment: Feudalism is not a common term world-wide. It would be good to define its meaning in this context.
Revision: The feudal university system refers to a rigid and hierarchical structure (Aubert et al. 2018), based on relations of abuse of power and bullying dynamics(Flecha, 2011).
Line 399-400
Comment: State why you have used the term upstander rather than bystander. (I think it is a good choice, but since bystander is more common in some places, an explanation would help.)
Revision: The term upstander is used here instead of bystander to highlight the active role of those who choose to intervene rather than remain passive observers (Puigvert et al., 2022).
Reviewer 2 Report
Comments and Suggestions for Authors
The authors offer an excellent review of the literature on behavioral and institutional changes regarding gender and sexual violence at Spanish universities. The authors offer an engaging discussion of the history of research, advocacy, new legislation and support for survivors and their supporters since the 2000s. The authors focus on two research questions. First, have the collective demands for positive change driven legislation and structural changes to increase safety? Second, have the new legislation and structural changes encouraged shifts in individual and institutional attitudes and behaviors related to accountability, support for survivors, denormalization of sexual and gender violence, and enhanced well-being in Spanish universities? The author’s meta-analysis of the literature and research suggest that significant behavioral and attitudinal shifts have occurred in Spanish universities and were driven by collective demands, new legislation, and structural changes. Examples of behavioral and attitudinal changes include, resisting codes of silence, increasing public discussion of the sensitive issue, reduction of fear in reporting and retaliation, and increased equity and meritocracy of professorship. This manuscript is well-written and powerful in terms of providing priceless knowledge that will inspire future research and collective action.
My only suggestion for improving the manuscript is for the authors to consider editing the title for clarification and brevity. The title is a bit confusing and does not do the manuscript justice. One idea: “Systematic Literature Review of the Research and Advocacy for Behavioral Change related to Denormalization of Gender Violence in Spanish Universities”.
I thank the authors for their excellent work on this important topic and look forward to seeing this article published.
Author Response
MARKS IN GREEN: Revised according to the highlighted similarity parts from the major revision.
- 299-302: We have decided to keep this part because it brings clarity to what is being explained. (Mark in yellow)
MARKS IN BLUE: Reviewers
Title
Comment : My only suggestion for improving the manuscript is for the authors to consider editing the title for clarification and brevity. The title is a bit confusing and does not do the manuscript justice. One idea: “Systematic Literature Review of the Research and Advocacy for Behavioral Change related to Denormalization of Gender Violence in Spanish Universities”.
Revision: Analysis and Review of the Research and Advocacy for Behavioral Change Related to the Denormalization of Gender Violence in Spanish Universities.
Reviewer 3 Report
Comments and Suggestions for Authors
The article is relevant and interesting as GBW is being addressed in universities worldwide as a result of various sporadic processes, some of which are addressed in the article: women reporting violence, university gender equality plan focusing on safety and inclusion, legislative changes, etc. The main focus of the article is on the behavioural changes at universities in Spain that were triggered by legislation passed in 2004, which set the stage for fundamental changes at universities in regard to preventing and investigating GBW.
The authors emphasise that universities are hierarchical (feudal) organisations with a strong power imbalance that allows for abuse towards both students and staff, which is changing with the government regulations that now make merit the main criterion for academic positions.
The main objective of this study is to analyse the changes in attitudes and behaviours in the university community, and two main hypotheses were put forward: The first was that social actors who have promoted change and safety from gender-based violence have influenced legislative changes, and the second was that legislative and structural changes implemented at universities have helped to change attitudes and behaviours toward gender-based violence.
The main method used to test the hypotheses was a literature review based on a final selection of 11 articles. The findings are more or less a chronology of change, describing the breaking of silence and the aftermath; how victims were empowered to speak out and act, and the increase in reporting of GBW.
The article's greatest strength is its systematic overview of the historical developments that enabled women to report and also compelled universities to take action against violence. It is also important to mention that the IGV definition was shaped by the Spanish movement against GBV at universities.
There are also some shortcomings that need to be addressed before publication. The cause-consequence relationship is week proven, as the change in normative actions does not yet produce a change in behaviour, which is also supported in the article with countermeasures of CRUE. The other point is that university hierarchies enable a position of power, whether it is based on merit or not. It is a power that comes with the hierarchical position that can lead to the abuse of that power, which is also proven by the case of Ana Vidu (here I suggest not using the full name of a student, as in this case the victim is known, but the name of the perpetrator has been anonymized – the same goes for the case of Mar Joanpere). The case was closed before it was investigated, which I assume was due to a position of power held by the perpetrator. It was reopened because of pressure from Harvard and not, as the authors wrongly assume, because the recognition and visibility of gender violence influenced UB's behaviour. The other point is that UB's behaviour was still in favour of the perpetrator, as it took them almost a year to send the case to the DA's office. So the fact that the case was reopened did not yet change their attitude and behaviour.
There is also a significant difference between the cases where the perpetrator is a student and the cases where the perpetrators are academic staff. It is therefore not surprising that the case against the student was decided in favour of the victim (lines 284-289)
The chronology shows how women’s groups and organisations in research and academia have developed an effective strategy for influencing legislation and changing university practise. It also shows that there were other processes besides legislative change that brought about the changing responses to GBW, e.g., the structure of degree programmes, the role of American universities, gender equality associations, etc.
The last point refers to the hypotheses that need to be tested. Putting them in the literature review is not methodologically correct as it is not possible to test them. I suggest replacing them with a good research question that allows for a more complex discussion of the results of the review.
Comments on the Quality of English Language
I am not a native speaker. By my oppinion there is a space for improvements.
Author Response
MARKS IN GREEN: Revised according to the highlighted similarity parts from the major revision.
- 299-302: We have decided to keep this part because it brings clarity to what is being explained. (Mark in yellow)
MARKS IN BLUE: Reviewers
- Line 507-513
Comment: There are also some shortcomings that need to be addressed before publication. The cause-consequence relationship is week proven, as the change in normative actions does not yet produce a change in behaviour, which is also supported in the article with countermeasures of CRUE.
Revision: confirm that as changes have occurred in the behavior of faculty, researchers, admin-istrative staff, and students, structural and legislative modifications in the field of gen-der-based violence have been successfully promoted, thereby improving safety in this environment. Moreover, these transformations have contributed to changing attitudes and behaviors, effectively denormalizing gender-based violence in the university setting. However, future research should further analyze the consequences of each of these changes in greater depth.
- Lines: 257; 288; 296: 354; 361
Comment: the case of Ana Vidu (here I suggest not using the full name of a student, as in this case the victim is known, but the name of the perpetrator has been anonymized – the same goes for the case of Mar Joanpere).
Revision:
- her student was being harassed (257)
- The victims (288)
- The victim (296)
- the victim’s case (361)
- the survivor (354)
- Lines: 86-89; 507
Comment: The last point refers to the hypotheses that need to be tested. Putting them in the literature review is not methodologically correct as it is not possible to test them. I suggest replacing them with a good research question that allows for a more complex discussion of the results of the review.
Revision:
- The research question is:
How have the demands of social actors influenced the formulation and implementation of legislative and structural changes on gender-based violence in universi-ties, and what impact have these changes had on attitudes and behaviors? (86-89)
- Responding to the research question (507)
- Lines: 263-266
Comment: The case was closed before it was investigated, which I assume was due to a position of power held by the perpetrator. It was reopened because of pressure from Harvard and not, as the authors wrongly assume, because the recognition and visibility of gender violence influenced UB's behaviour. The other point is that UB's behaviour was still in favour of the perpetrator, as it took them almost a year to send the case to the DA's office. So the fact that the case was reopened did not yet change their attitude and behaviour.
Revision: This demonstrates how external pressures, particularly from Harvard, influenced the behavior of the University of Barcelona, leading it to reconsider its position and reopen a case that was initially closed without being investigated.
Round 2
Reviewer 3 Report
Comments and Suggestions for Authors
I do not have any further comments.